# Pediatric Cardio-Oncology Medicine: A New Approach in Cardiovascular Care

**DOI:** 10.3390/children8121200

**Published:** 2021-12-18

**Authors:** Hugo R. Martinez, Gary S. Beasley, Jason F. Goldberg, Mohammed Absi, Kaitlin A. Ryan, Karine Guerrier, Vijaya M. Joshi, Jason N. Johnson, Cara E. Morin, Caitlin Hurley, Ronald Ray Morrison, Parul Rai, Jane S. Hankins, Michael W. Bishop, Brandon M. Triplett, Matthew J. Ehrhardt, Ching-Hon Pui, Hiroto Inaba, Jeffrey A. Towbin

**Affiliations:** 1Division of Pediatric Cardiology, The Heart Institute at Le Bonheur Children’s Hospital, University of Tennessee Health Science Center, Memphis, TN 38163, USA; gbeasle1@uthsc.edu (G.S.B.); jgoldbe4@uthsc.edu (J.F.G.); mabsi@uthsc.edu (M.A.); kbalduf1@uthsc.edu (K.A.R.); kguerrie@uthsc.edu (K.G.); vjoshi@uthsc.edu (V.M.J.); jjohn315@uthsc.edu (J.N.J.); jtowbin1@uthsc.edu (J.A.T.); 2Department of Diagnostic Imaging, St. Jude Children’s Research Hospital, Memphis, TN 38105, USA; cara.morin@stjude.org; 3Division of Critical Care Medicine, St. Jude Children’s Research Hospital, Memphis, TN 38105, USA; Caitlin.Hurley@STJUDE.ORG (C.H.); Ray.Morrison@stjude.org (R.R.M.); 4Department of Bone Marrow Transplantation & Cellular Therapy, St. Jude Children’s Research Hospital, Memphis, TN 38105, USA; brandon.triplett@stjude.org; 5Department of Hematology, St. Jude Children’s Research Hospital, Memphis, TN 38105, USA; Parul.Rai@STJUDE.org (P.R.); jane.hankins@stjude.org (J.S.H.); 6Division of Solid Tumor, St. Jude Children’s Research Hospital, Memphis, TN 38105, USA; Michael.bishop@stjude.org; 7Division of Cancer Survivorship, Department of Oncology, St. Jude Children’s Research Hospital, Memphis, TN 38105, USA; matt.ehrhardt@stjude.org; 8Department of Epidemiology and Cancer Control, St. Jude Children’s Research Hospital, Memphis, TN 38105, USA; 9Division of Leukemia/Lymphoma, St. Jude Children’s Research Hospital, Memphis, TN 38105, USA; ching-hon.pui@stjude.org (C.-H.P.); hiroto.inaba@stjude.org (H.I.)

**Keywords:** pediatric cardiology-oncology, early detection, cardiovascular healthcare, oncology therapies, individualization of therapies, predictive healthcare models

## Abstract

Survival for pediatric patients diagnosed with cancer has improved significantly. This achievement has been made possible due to new treatment modalities and the incorporation of a systematic multidisciplinary approach for supportive care. Understanding the distinctive cardiovascular characteristics of children undergoing cancer therapies has set the underpinnings to provide comprehensive care before, during, and after the management of cancer. Nonetheless, we acknowledge the challenge to understand the rapid expansion of oncology disciplines. The limited guidelines in pediatric cardio-oncology have motivated us to develop risk-stratification systems to institute surveillance and therapeutic support for this patient population. Here, we describe a collaborative approach to provide wide-ranging cardiovascular care to children and young adults with oncology diseases. Promoting collaboration in pediatric cardio-oncology medicine will ultimately provide excellent quality of care for future generations of patients.

## 1. Introduction

There has been a significant improvement in the care and survival of pediatric patients diagnosed with cancer (e.g., leukemia/lymphoma and solid tumor) [1]. While previous studies have associated cardiotoxicity in children treated with anthracycline, alkylating agents, radiation, and immune checkpoint inhibitors, emerging new therapies have potentially harmful consequences for the cardiovascular (CV) system; thus, the implications of cancer therapy-related cardiovascular dysfunction (CTRCD) remain to be completely elucidated [2,3]. An estimated 50% of childhood cancer survivors harbor some degree of subclinical CTRCD. Therefore, timely diagnosis of asymptomatic disease is an opportunity not only to extend short- and long-term survival rates but also to improve the quality of life of this vulnerable population [4,5].

In 2016, the Heart Institute at Le Bonheur Children’s Hospital (LBCH) began to expand relationships with clinical specialties at St. Jude Children’s Research Hospital (SJCRH) to deliver comprehensive CV healthcare, institute academic pathways, and advance research collaboration to improve outcomes in children and young adults undergoing cancer and hematology treatments. The pediatric cardio-oncology (PedCO) team comprises a set of consulting cardiologists with expertise in myocardial disease. The “front-line” team is present during daily medical rounds, identifying, evaluating, and optimizing therapy for patients with clinical and subclinical CTRCD. An in-house PedCO team allows for shifting the paradigm from managing manifested cardiovascular illness to preventing and identifying subclinical disease. As a consultant service, the PedCO members provide day-to-day recommendations to oncologists in charge of the well-being of patients with cancer. In addition, the PedCO staff is well supported year-round by other cardiovascular subspecialists with additional expertise in dysrhythmias, imaging, surgery, and interventional cardiology, (Figure 1).

The purpose of this review is to outline the multidisciplinary collaboration between the cardiovascular services from LBCH and the oncology/hematology providers at SJCRH to achieve an exceptional clinical and academic practice for children with co-existing cardiovascular and oncology disease.

## 2. Evolution of Pediatric Cardio-Oncology as a Discipline

The initial steps in recognizing myocardial toxicity following cancer therapies were established in the 1960s after the incorporation of anthracyclines in the management of cancer. The association of these drugs with cardiac dysfunction has influenced the establishment of regulatory policies for drug administration. Until the early 1990s, the Cardiology Committee of the Children’s Cancer Study Group formulated pediatric-specific recommendations for imaging monitoring of children undergoing chemotherapy [6]. Since then, there has been an upsurge of scientific statements and research venues to manage cardiotoxicity, including the incorporation of advanced imaging modalities, medications, and serum biomarkers for the stratification of cancer patients at risk of developing a cardiac illness. In recent years, clinical guidelines in cardio-oncology have sparked initiatives to shift the health system from one based on the management of diseases to one focused on prevention and wellness.

## 3. Surveillance and Management

The institution of surveillance schemes to detect CV adversity in oncology treatments has been highly prioritized, as cardiotoxicity encompasses significant morbidity in this patient population [7]. The groundwork of surveillance includes the identification of comorbidities and predisposing risks by obtaining a complete personal and family history; the incorporation of serial reproducible cardiac imaging; and the establishment of baseline and longitudinal profiles including physical examination, vital signs, and cardiac biomarkers before, during, and after oncology treatment/planning (Figure 2).

### 3.1. Screening

Screening aims to identify patients at risk or those harboring any CV illness before the initiation of or alongside oncology therapy. The risk of developing CTRCD may be determined by the nature of cancer, the domains associated with the therapy, patient-related comorbidities, CV reserve at baseline, familial CV traits, and suboptimal function in other organs and systems [8]. Given the wide variety of therapeutic options (e.g., chemotherapy, immunotherapy, radiation, stem cell transplantation, surgery, and molecularly targeted therapy), it is difficult to consolidate single guidance in the frequency of screening and surveillance for CTRCD. The institutional risk assessment at SJCRH is determined by independent oncology protocols based on the factors noted above. For patients harboring comorbidities or those patients who will receive cancer therapy with potential cardiovascular toxicity, screening typically includes CV imaging (echocardiography (echo) or cardiac magnetic resonance imaging (cMRI)), electrocardiography, a complete metabolic panel, and in many cases, baseline cardiac biomarkers, such as troponins and N-terminal pro B-type natriuretic peptide (NT-proBNP). In addition to a baseline evaluation, longitudinal screening is advised for all patients receiving cancer therapy [3]. Long-term evaluations provide valuable data to compare clinical and subclinical changes of CTRCD (Table 1). Throughout these evaluations, patients may encounter signs and symptoms of undesirable adverse effects. However, the extent to which cardiac dysfunction constitutes a contraindication for cancer therapy is ill-defined. For this reason, the standardization of screening throughout therapy has allowed us to identify potentially reversible CTRCD, institute timely CV management, and initiate collaborative efforts to balance the risk of toxicity of conventional or alternative cancer therapies [4]. As such, our PedCO approach has generally supported and accommodated rather than prohibited treatment plans from oncology services.

### 3.2. Diagnosis

Traditionally, cardiotoxicity is diagnosed via echo when there is (1) a 10% decrease in LV ejection fraction (LVEF), (2) an LVEF with an absolute value <55% and/or an abnormally low shortening fractioning (SF) [6]. Using LVEF as the sole marker of ventricular dysfunction is the suboptimal delineation of cardiac remodeling, a preclinical stage of myocardial injury, inflammation, and fibrosis leading to altered mass, geometry, and function of the heart [15]. This preclinical stage of myocardial remodeling is not detected by echo. Similarly, these patients frequently encounter a variety of load effects, such as systemic and pulmonary hypertension, sepsis, pericardial effusions, and hyperhydration; these factors cause variability in the calculation of LVEF [16]. To balance these challenges, our PedCO team has recommended moving away from the utilization of 2D echocardiography (such as SF) and incorporating newer imaging modalities, such as 3D echocardiography, strain analysis, and cMRI. This topic is of importance because the combination of using newer imaging tools and the daily presence of a cardiology team has allowed us a valuable diagnostic opportunity to identify previously under-recognized cardiovascular injuries that hinder appropriate oncologic management (Figure 3).

Such pathologies include heart failure with preserved ejection fraction, a form of cardiac failure associated with diastolic dysfunction, which has been diagnosed more frequently than systolic dysfunction at our institution in a cross-sectional observation (Table 2). Timely diagnosis of diastolic dysfunction is key to avoid progression into arrhythmias, cardio-pulmonary mechanic deterioration, and myocardial fibrosis.

Another underlying cause of CTRCD in the pediatric population includes vascular toxicity. Targeting endothelial factors as an anticancer therapy, the utilization of medications affecting vasomotor tone, the administration of high-dose steroids, and the frequent surge of catecholamines in these patients contribute to the development of endothelial dysfunction and vascular hypercontractility; these conditions manifest in vascular hypertension, nephropathy, and coronary artery disease [17,18]. Similarly, the presence of intracardiac, venous, and arterial thromboembolic phenomena has been one of the most common reasons to consult our PedCO service (Table 2). These issues are not unexpected because patients with cancer harbor many factors contributing to increased pro-thrombotic risks, such as hypercoagulability (increased circulating procoagulant factors or decreased antifibrinolytic factors), increased stasis (immobility and blood hyperviscosity), and endothelial injuries [19,20].

In contrast to the traditional definition of CTRCD (e.g., an incident 10% decline in ejection fraction) [21,22], our PedCO team has adopted a systematic approach to the diagnostic classification of cardiovascular dysfunction based on the American College of Cardiology/American Heart Association (ACC/AHA) staging system for heart failure [23]. In this system, Stage A includes patients at risk of cardiovascular toxicity, Stage B includes patients with evidence of cardiotoxicity in a pre-clinical or asymptomatic phase, Stage C includes patients with mild to moderate cardiovascular symptoms associated with CTRCD, and Stage D includes individuals requiring hospital-based CV support (Table 3).

### 3.3. Prevention

Prevention of CTRCD begins at cancer diagnosis, as early recognition and mitigation of adverse effects are paramount to counteract intermediate and long-term CV sequelae.

A broad approach to strategize preventive and therapeutic options includes the classification of patients into two main groups: patients deemed to receive primary prevention (PedCO Stage A) and those with manifested CTRCD requiring secondary-type therapies (PedCO Stages B, C, and D) (Table 3).

Our primary prevention strategies begin with a comprehensive assessment of comorbid risk factors, providing education and counseling regarding strategies to mitigate these factors where appropriate (e.g., emphasizing the importance of lipid, weight, and diabetes management; physical and nutritional rehabilitation). Collectively, dietitians, nurses, physical therapists, pharmacists, social workers, physicians, and other associated health providers contribute to this multidisciplinary messaging.

Where applicable, our team proactively recommends the use of alternative anthracyclines formulations, such as the liposomal presentation. The liposomal formulation has been shown to reduce cardiotoxic effects secondary to the size of the chemical components, which limits diffusion through the endothelial lining of the myocardial microvasculature [24,25]. An additional recommendation includes the use of slow infusion rates rather than a bolus administration of anthracyclines to diminish the accumulation in the myocardium. However, this recommendation requires a multidisciplinary conversation, as this approach may be counterbalanced by more discomfort due to prolonged hospitalization [24,26,27].

Dexrazoxane is a chelating agent that decreases the production of iron-induced free radicals and induces the degradation of topoisomerase IIb to reduce anthracycline-associated myocardial and endothelial injury. To date, dexrazoxane is the only agent approved by the United States Food and Drug Administration for preventing anthracycline-related cardiomyopathy [22]. Dexrazoxane has been approved for use as a cardioprotectant agent in women with metastatic breast cancer; while long-term benefits and children remain uncertain, a growing body of literature suggests short-term benefits in pediatric patients [28,29]. Its use has therefore been routinely incorporated into multiple institutional treatment protocols, particularly for those expected to receive cumulative doxorubicin equivalent anthracycline doses of ≥250 mg/m^2^ [30,31], a threshold shown in numerous studies to be associated with the greatest risk for subsequent cardiotoxicity [32,33,34,35,36]. Where feasible, the primary oncology team may consider the substitution of anthracyclines for alternative, less cardiotoxic agents.

### 3.4. Treatment

Secondary prevention strategies incorporate continuum screening options, largely dictated by individual treatment protocols for children undergoing active cancer treatment or in early follow-up after completion of therapy. The evolving incorporation of cardio-oncology has played a significant role in mitigating the progression of CV disease by instituting timely and effective therapeutic options, thus providing a balance between anti-cancer efficacy and CTRCD [37,38,39]. However, at the time of this publication, there exist no guidelines for the management of all the underlying factors associated with CTRCD in children (Figure 3). Therefore, we derive our therapeutic approach from the heart-failure guidelines from the ACC/AHA [23] (Table 3). As previously shown, early institution of medical therapy has provided benefit in the general population to reverse or ameliorate cardiovascular disease by inhibiting neurohormonal and hemodynamic maladaptation [40,41].

Diastolic dysfunction (DD), described as an early echocardiographic finding before systolic dysfunction, is detected in children receiving anthracyclines and radiation [42,43]. Contributing factors to this phenomenon include myocardial damage from the cancer therapy triggering abnormal thickening of the ventricular walls, decreased elasticity from myocardial loss and fibrosis, and endothelial dysfunction compromising myocardial perfusion and promoting systemic arterial hypertension [42,44]. Studies have also shown that early and subclinical stages of diastolic dysfunction trigger a compensatory elevation of NT-proBNP (promoting cardiac remodeling and altered myocardial geometry) and heart rates (requiring higher myocardial energetics and oxygen consumption) [45]. Based on this evidence, beta-blockade therapy may be considered to avoid the progression into a more serious stage. However, while considering beta-blockers to modulate heart rates, inadequate myocardial relaxation, and systemic arterial hypertension, one should take into account the presence of demanding hemodynamic states due to inflammation, infection, anemia, endothelial dysfunction, and inappropriate catabolism [42,45,46].

Left ventricular systolic dysfunction has been extensively investigated and has been linked to suboptimal outcomes while undergoing cancer management [21,47]. In children, EF% and strain analysis remain the surrogate markers to evaluate ventricular dysfunction [3,14,48,49]. Global longitudinal strain has been reported to be the most efficient strain parameter to quantify left ventricular mechanics. In our experience, it has been helpful to identify focal and global ventricular dysfunction although we recommend reviewing vendor-specific values of normal/abnormal when using it clinically [14,48,50]. When the ventricular systolic function is suboptimal at the regional or global levels, the use of beta-blockers (BB), angiotensin-converting-enzyme inhibitors (ACEIs), and angiotensin receptor blockers (ARBs) has been associated with cardio protection [3,21,51,52]. Regardless of the cardiotoxic insult, reduced afterload stress, inappropriate tachycardia, and conservation of oxygen consumption/myocardial energetics should be addressed immediately (PedCO stage B to D). Medical therapy for compensated systolic heart failure with ACEi and ARBs is recommended to initiated at a low dose and uptitrated to the recommended target dosage based on tolerance [53,54]. In this population, the risk of renal dysfunction, hyperkalemia, hypotension, and drug-drug interaction should be frequently monitored.

Our recent practice has also included the use of sacubitril/valsartan (angiotensin receptor blocker/neprilysin inhibitor), as it has been shown to improve morbidity and mortality in adults with heart failure compared to ACE inhibition, with similar observations emerging in pediatrics [52,55].

A unique consideration to this patient population includes the ICU admission for cardiac optimization, for example, patients with chronic and stable cardiac dysfunction who are typically well managed on oral therapy but who require a more robust cardiac function to meet chemotherapy and/or a hematopoietic cell transplantation criterion. For these patients, an accelerated optimization of cardiac function (via a short-term milrinone infusion) provides them the opportunity to proceed with their cancer therapies and avoid disease relapse. This scenario provides an example of patient-centered decision making in this population. That is, all patients are recommended to undergo a baseline CV screening; CV treatments are guided by the nature and severity of the diagnosis, to prevent further CV deterioration, and to achieve completion of oncologic therapies.

Cancer and end-stage heart failure should allow thoughtful and timely consideration of advanced therapies in stage D. The use of mechanical circulatory support (MCS) has expanded in recent years and is widely used in advanced heart failure [56]. From our perspective, the decision for the use of MCS warrants careful consideration of certain factors, such as reversibility of underlying cardiac failure, oncologic prognosis, sepsis, cerebral edema, cytokine storm, vasoplegia, and clinical conditions associated with increased risks of bleeding/thrombosis.

Arterial hypertension (pulmonary and systemic) has been identified as a significant cause of morbidity and mortality; it has been implicated in strokes, coronary heart disease, peripheral arterial disease, heart failure, and renal disease [57,58]. Given the plethora of cancer-therapy agents and underlying mechanisms causing arterial hypertension, a multidisciplinary approach is advised. With no trials comparing antihypertensives in these patients, hypertension in children with cancer may be managed initially based on current guidelines [59].

As previously stated, children with cancer are a frail population with multiple comorbidities who require a variety of providers and medications, which may increase the risk of drug interactions and associated adverse drug events. Effective interventions to improve the effects of polypharmacy involve the inclusion of pharmacy staff members, the utilization of electronic medical records, and the use of drug-interaction screening via software programs [60].

## 4. Pediatric Cardio-Oncology Healthcare Model

### 4.1. Inpatient PedCO

Inpatient service is one of the most relevant aspects of our CV care model. Since 2016, the primary cardio-oncology service has been gradually integrated six pediatric cardiologists with expertise in myocardial disease who provide continuous evaluation and management of CTRCD. This “front-line” cardio-oncology group has been supported by additional cardiology and oncology stakeholders who are available to deploy further expertise for the therapeutic framework (Figure 1).

### 4.2. Outpatient PedCO

The implementation of an outpatient PedCO practice is important because children who have received anti-cancer treatments require frequent follow-up during and after therapy. Of primary importance is the ready availability of pediatric cardiologists who have familiarity with the spectrum of cancer-directed therapies and their associated cardiotoxicities. In addition, each clinic is staffed by a registered nurse who serves as the primary point of patient contact during and between visits and an experienced laboratory equipped to provide echocardiograms, electrocardiograms, and ambulatory monitoring. Cardiopulmonary stress testing and cMRI are available on-site and can be integrated when appropriate. The frequency of clinic visits and imaging evaluations depends on both the severity of cardiac dysfunction and the timing and nature of the oncology therapies (Table 3).

### 4.3. Pediatric Oncology-Critical Care

SJCRH is a pioneer in the emerging subfield of pediatric oncology-critical care. Disease processes and interventions in critical illness in the oncology/transplant population are distinctive and require specialized care. The Division of Pediatric Critical Care Medicine at SJCRH has received national recognition for clinical excellence in the treatment of critically ill children with cancer.

Common cardiac diagnoses requiring ICU management include pericardial effusions with pre-tamponade or tamponade physiology, pulmonary and systemic arterial hypertension, and myocardial dysfunction (systolic and/or diastolic) from a variety of etiologies, including chemotherapy, infective and chemical myocarditis, pulmonary thromboembolism, and sepsis. The critical care team contributes to a multidisciplinary approach in individuals harboring hyperinflammatory conditions, such as cytokine release syndrome after chimeric antigen receptor T-cell therapy and hemophagocytic lymphohistiocytosis. Critical care for cardiac dysfunction is designed based on the hemodynamic findings, imaging studies, and functional classification of the patients (Table 3). We have observed that hemodynamic instability often presents with a mixed shock, cardiogenic and distributive, due to CTRCD, sepsis, and other hyperinflammatory states.

### 4.4. Congenital Heart Disease

Congenital heart disease (CHD) affects approximately 1 in 100 live births [61], and children with CHD have a higher rate of childhood cancer than individuals born without CHD [62]. Leukemia and lymphoma are the most common pediatric cancers observed in children with CHD, each accounting for approximately 28% of cases [62]. For instance, an estimated 40–60% of patients with trisomy 21 harbor CHD, and this group is at increased risk for developing transient myeloproliferative disorder (exclusively seen in patients with Down syndrome), acute megakaryoblastic leukemia (500-fold), and acute lymphoblastic leukemia (10- to 20-fold) compared to the non-Down syndrome population [63,64,65]. Similarly, 22q11.2 deletion syndrome is a relatively frequent multisystem disorder with phenotypic characteristics, including CHD (interrupted aortic arch, truncus arteriosus, and tetralogy of Fallot) and increased risk of lymphomas and thrombocytopenia [66]. An analogous group of patients is represented by the disorders grouped into the category of RASopathies (abnormalities in the RAS–mitogen-activated protein kinase pathway). For instance, up to 80% of these patients have CHD (pulmonary valve stenosis and atrioventricular canal defects) with an approximately 8-fold increase in the development of myeloproliferative and/or solid tumors [65,67]. For these reasons, we have implemented a dedicated clinic at SJCRH that provides the expertise of managing CHD.

### 4.5. Cardiac Electrophysiology

Incorporating a multidisciplinary pediatric cardiology team in the management of patients with cancer disorders aims to offer comprehensive healthcare based on collective expertise before, during, and after oncology therapies (Figure 1). Cardiac dysrhythmias represent a subtype of cardiotoxicity that may occur with or without cardiac dysfunction. Deviation from the normal rate and/or rhythm can be attributed to variations in impulse formation or conduction disturbances. Generally, a decrease in activity results in bradycardia, while an increase in activity results in tachycardia. Failure of impulse formation, such as in sinus pause or conduction block, as seen in AV nodal block, contributes to bradyarrhythmia. In contrast, reentrant excitation enhances automaticity, and triggered activity results in tachyarrhythmia. Cancer therapy-induced arrhythmias are an important form of cardiotoxicity that may develop as an acute or late toxicity and have significant implications for overall morbidity and mortality [68]. Arrhythmias result primarily from therapy-induced modification of molecular pathways critical to arrhythmia genesis or secondary to cardiac tissue damage that leads to the development of arrhythmogenic substrates [69]. Moreover, supraventricular tachycardia has been associated with cytotoxic chemotherapies, such as anthracycline, cyclophosphamide, cisplatin, and alkylating agents [3,10,69,70,71]. Ventricular arrhythmias, particularly those associated with QTc prolongation, have been associated with kinase inhibitors, arsenic trioxide, and anthracyclines [72]. QTc prolongation requires careful monitoring, given its associated risk of torsade de pointes [73]. QTc prolongation is also common in this population due to the concomitant use of QT-prolonging medications, such as those for controlling nausea and emesis. With numerous medications recognized to have the potential to prolong the QTc interval, CredibleMeds.org is a website offering a list of more than 200 medications grouped into four risk-categories based on their association with QTc prolongation [74].

Corticosteroid and anti-microtubule drugs may induce bradycardia and conduction abnormalities, occurring in up to 25% of patients [70,75]. Additionally, spinal and thoracic irradiation reportedly cause the abnormal presence of Q waves, ST-segment changes, and decreased QRS voltage [76].

Pre-therapy electrocardiography assessment allows the identification of pre-existing conduction or repolarization abnormalities and provides baseline measurements for future serial comparisons. This assessment also serves as a diagnostic tool for monitoring CV diseases during and after cancer therapy, risk factors in clinical trials, and adverse events in the longitudinal trajectory of drug development. We also provide routine ambulatory monitoring in patients with an increased risk of arrhythmia or conduction disease. Ambulatory monitoring may provide additional information in symptomatic patients, allowing us to discern between normal versus abnormal rhythms associated with reported symptoms. Long-term monitoring with an implantable loop recorder (ILR) has also been useful for intermittent symptoms or episodes that have not been captured by standard ambulatory monitors. ILR provides clinically useful information for pediatric patients within six months of implantation [77]. Collaboration with our cardiac electrophysiologists has been a key component in patient well-being by decreasing cardiac rhythm-associated morbidity and mortality in this population.

### 4.6. Sickle Cell Disease (SCD)

In patients with SCD, the lifespan remains at the median survival of ~48.0 years (95% CI: 44–58) [78]. This early mortality is strongly associated with concomitant end-organ damage that begins in childhood [79,80,81]. Furthermore, cardiopulmonary causes of death account for approximately 60% of all premature deaths in adults [82]. Cardiac manifestations are heterogeneous, and they are dependent on age, genetics, the quality and quantity of hemoglobin, and therapeutic options. We have observed pathological CV manifestations during childhood and adolescence, including persistent high cardiac output states, eccentric and concentric left ventricular hypertrophy, scattered foci of fibrosis, atrial enlargement, and systolic and diastolic dysfunction. These findings have been correlated with previous observations [82,83,84]. Therefore, we believe that early intervention may prevent or reduce cardiac complications. As a result, we provide systematic CV assessments, which collect high-quality data from CV screening evaluations in adolescents with SCD. This conceptual model integrates research into clinical care, analyzes the CV effects of modifying therapy (hydroxyurea and blood transfusions) and newer therapeutic agents, and incorporates advanced CV imaging and serum biomarkers to better elucidate the natural progression of cardiac injury.

### 4.7. Cardiovascular Imaging

Providing a detailed CV visualization is essential for diagnostic workup, longitudinal surveillance, and guidance of medical management. Echocardiography remains the most widely used imaging modality for cardiac functional and structural assessments in children undergoing cancer therapy [3]. This is in part due to its ready availability and lack of radiation [85]. For most contemporary oncology experts, the LVEF remains the gold standard marker to guide therapeutic approaches; fortunately, the use of fractional shortening to assess LV systolic function (LVSF) is disappearing from cancer protocols. It is important to note that these patients frequently encounter cardiac preload and afterload factors affecting the ventricular mechanical efficiency, which biases the calculation of a given LVEF or SF. Some of these factors may include systemic arterial hypertension, rapid shifts of intravascular volumes, and tachycardia [13,86]. To guide clinical judgment from echocardiography reports, our front-liner team is present during medical rounds and consults to elaborate on clinical interpretation from CV imaging. Frequent inquiries to be addressed by echocardiography include the assessment of pulmonary pressures, the presence of thromboembolic phenomena, pericardial effusions, and the scope of ventricular function. Patients undergoing cancer therapies can experience heart failure, but this manifestation may represent just the tip of an iceberg comprising a wide spectrum of clinical and subclinical adverse effects (Figure 3).

*Systole*: With the interobserver differences in measuring EF, especially in small children, we have built-in more precise measurement techniques, such as 3D echocardiography. This imaging modality has the ability to be semi-automated by artificial intelligence, which has been an important way to reduce miscalculations and avoid the serious limitations of 2D echocardiography and LVSF [87]. Locally, the calculation of the 3D LVEF is the preferred method for the evaluation of systolic function because it is considered to have decreased interobserver variability and greater reproducibility [88]. When 3D echocardiography is not possible, 2D echocardiography or Simpson’s biplane method becomes the default source of information to calculate the LVEF.

*Diastole*: Abnormal diastology is one of the main reasons to consult the PedCO team (Table 3). Anecdotally, this phenomenon has been more evident in patients undergoing hematopoietic cell transplantation and immune-modulatory therapies. While echocardiographic diastology may not have high sensitivity in young children, we believe diastolic function is best assessed when followed serially. Our echocardiography laboratory follows the recommendations from the American Society of Echocardiography to evaluate diastolic dysfunction [89]. In addition, we have been able to integrate Z-score nomograms for age, recognizing that adult standards of grading diastolic function are not applicable [89,90,91]. The parameters used to evaluate diastolic function (transmitral inflow profile, mitral annular velocities by tissue Doppler, the tricuspid regurgitation velocity, and the dimension of the left atrial volume) should aim to help clinicians to answer the following questions: (1) Is diastolic function normal, abnormal, or indeterminate? (2) Is the left ventricular filling pressure elevated or normal?

The role of cMRI stands out for its accuracy, reproducibility, and lack of ionizing radiation [92]. In addition to improved accuracy compared to echo, cMRI has multiple additional advantages, including: (1) the ability to provide myocardial tissue characteristics such as early signs of edema, inflammation, and iron deposition; (2) changes in myocardial mass/volumes (atrophy); (3) characterization of ischemic and non-ischemic patterns of fibrosis; (4) global and segmental quantification of systolic function; and (5) delineation of arterial, pericardial, and valvular disease [93]. At SJCRH, the utilization of cMRI has become more common for the screening and diagnosis of CTRCD. cMRI has the additional advantage of improved right ventricular assessment compared to echo. Additionally, some patients (e.g., post chest radiation) have poor acoustic windows for echo, which is not an issue for cMRI. As the gold standard imaging modality for cardiac function, cMRI is also used when there is discordant echocardiographic data.

Given the nature of the unique patient population at SJCRH, patients do not follow one single imaging schedule/protocol. Rather, the initial screening and longitudinal surveillance are provided by independent protocols based on the underlying disease, the nature of the therapies, the individual history of CV disease, and comorbidities. When clinical or subclinical CV pathology is apparent, then imaging and cardiovascular consultations are provided more frequently.

### 4.8. Long-Term Follow-Up

Recognition of late-occurring health complications in aging childhood cancer survivors has led to the establishment of long-term follow-up cohorts that have proven essential to the identification of clear associations between specific cancer treatments and late-occurring chronic health conditions [94,95,96]. Among the most well-recognized are late cardiovascular toxicities, which become the leading cause of noncancer-related death as early as 30 years from childhood cancer diagnosis [97]. Left ventricular systolic dysfunction is of particular interest and has been associated with prior anthracycline chemotherapy and chest radiation exposure [33,36,98]. Growing understanding of these associations prompted the establishment of cancer survivorship as a unique discipline and a priority in the cancer care continuum [99] as well as the development of several long-term follow-up guidelines to assist providers in the early recognition of these late effects [100,101,102,103,104,105]. In the United States, the COG Long-Term Follow-Up Guidelines are the most frequently utilized [102,106]. For cardiovascular toxicity, these guidelines outline strategies for screening, both for cardiotoxicity (e.g., echocardiography for cardiomyopathy) and for comorbid conditions that increase the risk of cardiotoxicity (e.g., hypertension). Multiple studies have demonstrated the cost-effectiveness of screening for cardiotoxicity, particularly for individuals at high risk for cardiomyopathy, thus establishing screening as a standard of care in at-risk individuals [107,108].

Our typical practice at SJCRH involves transitioning survivors of childhood cancer to an onsite, dedicated cancer survivorship clinic (the After Completion of Therapy [ACT] clinic), beginning approximately five years from remission, where they are seen annually by a team of advanced practice providers and physicians trained in family medicine, internal medicine, pediatrics, and pediatric oncology for 10 years from a cancer diagnosis or until age 18 years, whichever occurs later. For cardiovascular screening, the ACT clinic adheres to COG guidelines [102,106]. Many ACT patients are seen in conjunction with the PedCO team for early and/or progressive cardiac dysfunction and undergo a proactive and collaborative transition to community cardiologists for ongoing care. Because of the average age of survivors at this transition, the preference is typically to identify an adult cardiologist when appropriate to minimize subsequent transitions. While it may be difficult to ensure that each PedCO patient transitions to appropriate adult healthcare, discussing such care should take place at all adolescent visits in the ACT and PedCO clinics. Fortunately, given the increased recognition of adult cardio-oncology specialists, such physicians are increasingly available in most metropolitan areas [109].

## 5. Academic Practices

In the United States, SJCRH established the first pediatric onco-critical care fellowship, a one-year training program for graduates of pediatric critical care medicine fellowships who wish to gain further education in this discipline. Trainees spend considerable time in specialized ICUs and rotate with inpatient teams on hematopoietic stem cell transplantation and chemotherapy services as well as in the acute care clinic. This approach provides a foundation for the diagnosis, treatment, and supportive care of patients with cancer. While we anticipate that the regional impact of this program will be substantial with regards to developing well-trained practitioners, we recognize that the global impact of this program will be limited. The Department of Global Pediatric Medicine at SJCRH estimates that 80% of children with cancer live in low- and middle-income countries [110]. Therefore, this department has integrated regional and global investigational and educational initiatives directed at healthcare providers to build self-maintenance while integrating research and health services adjusted to local capacity and needs. Our group recognizes the need to address PedCO at this global level and therefore has integrated educational activities in various platforms of the Global Pediatric Medicine Forum, such as the annual Pediatric Onco-Critical Care Symposium and the Pediatric Onco-Infectious Disease Symposium. Additionally, our recently established Pediatric Cardio-Oncology webinar hosted by the Heart Institute at LBCH, SJCRH, and the University of Tennessee Health Science Center has been endorsed by the International Society of Cardio-Oncology (https://ic-os.org, accessed on: 1 October 2021) and has reached more than 40 countries after the third bi-monthly presentation (Figure 4).

## 6. Discussion

Improved pediatric cancer survival rates have been achieved in the last decades mainly due to an increasing number of novel therapeutic options and the incorporation of a systematic multidisciplinary approach for supportive care. With the rapid development of anti-cancer modalities, early CV adversities, and long-term sequelae from some therapies, there has been a demand to propel the evolution of PedCO. From our global interaction with other pediatric health specialists caring for children with cancer, we have sensed a paucity of evidence-based information regarding pediatric CTRCD. This phenomenon is likely a consequence of multiple factors. First, there is no standardized definition of subclinical CV injury caused by oncology therapies. Second, most oncology protocols do not implement advanced imaging modalities or comprehensive biomarker assays for the surveillance of adverse effects. Third, genetic predispositions to CTRCD remain understudied. Finally, there has been an incomplete understanding of the effects of CV on benign hematological diseases, such as SCD, that are known to cause chronic organ damage, resulting in increased morbidity and mortality in children [111,112,113,114].

Even though the information regarding the operational components and infrastructure to support this discipline in pediatric programs remains scarce, herein, we unveiled the successful approach from the Heart Institute at LBCH to deliver wide-ranging cardiovascular care to children and young adults at SJCRH. Some of the key elements for this success have included:

(1) Medical care. CTRCD is one of the main determinants of morbidity and mortality in patients with cancer. Thus, incorporating a “front-line” team with expertise and interest in cardio-oncology has been able to promote a shift towards greater awareness of the cardiovascular adversities from cancer and cancer therapies (Figure 2). The clinical utility of including a dedicated cardiology service at a pediatric cancer center has been reflected by the increased rates of screening, prevention, and adequate management of CTRCD. We have also implemented an action plan to simplify the CV care (prevention, early diagnosis, and satisfactory management) of PedCO patients (Table 3). This staging system uses letters A to D to allow for early recognition of children who are at risk of developing CTRCD and identifies disease progression to help decide the most appropriate treatment options.

(2) Opening communication between multiple stakeholders. Establishing a multidisciplinary cardio-oncology approach has offered comprehensive healthcare based on collective expertise before, during, and after oncology therapies, as shown in Figure 1. We have found that opening venues of communication in a multidisciplinary forum provides a valuable way to practice and transform the outcome in a positive way. Interactive platforms can help the stakeholders to identify patients at risk of CTRCD, discuss patient care to arrange ways to mitigate CV disease, and understand the rationale behind cancer and/or cardiology therapies. In addition to promoting educational pathways and investigational collaboration, multidisciplinary podia with inclusion of patients and caregivers allow for the identification of shared decision-making goals.

(3) Administrative support. The initiation of a PedCO program may be complex. This element has been counterbalanced by bringing together cardiologists and oncologists to express the need for a PedCO service to institutional leaders and managers. Although there is a relative paucity of outcome data related to PedCO services, our cardiology team has enabled optimal care of children with cancer and has been competent to shift resources to prevent cardiotoxicities and their associated costs.

(4) Educational pathways. As the field of oncology advances the survival rates for all children, there is an increasing demand to optimize CV health in this population. Fortunately, in recent years, this demand has increased the number of dedicated cardio-oncologists worldwide. The future of optimal care in this population will depend on the development of training pathways to guarantee excellence in the quality of care for future generations of patients. Our Heart Institute is committed to advance the PedCO field.

Within the future directions of this field, research creativity should seek to leverage data to drive innovative improvements in CV health, quality of life, and cost-effective health systems. Investigational studies should explore the incorporation of novel biomarker assays to evaluate the spectrum of CTRCD from drug trials. These assays should also facilitate confident guidance in the management of CV disease. The incorporation of advanced cardiac imaging such as cMRI could integrate superior characterization of the presence of clinical and subclinical CTRCD in research trials and clinical practice. Sharing databases for the domains of multi-omics and artificial intelligence would bring rapid advances in research by allowing investigators to manage large datasets. These systems have also contributed to the application of genotype-phenotype coupling, pharmacogenetic-pharmacogenomic profiles, and individualization of therapies based on algorithms using predictive models in many other health systems.

In summary, PedCO comprises cardiologists who provide advice to patients, caregivers, and other health providers in decision-making processes to optimize the care of children with cancer. Forming an interdisciplinary cardiology team has collectively reduced knowledge gaps and improved the outcomes of children with cancer. We have also noted that early detection of CV diseases and pre-emptive initiation of medical management improves the quality of life as well as short- and long-term survival rates in children and young adults with cancer and cancer survivors. Therefore, we continue to recommend anticipatory medical practices to transform the CTRCD paradigm of illness management to prevention and wellness.

## Figures and Tables

**Figure 1 children-08-01200-f001:**
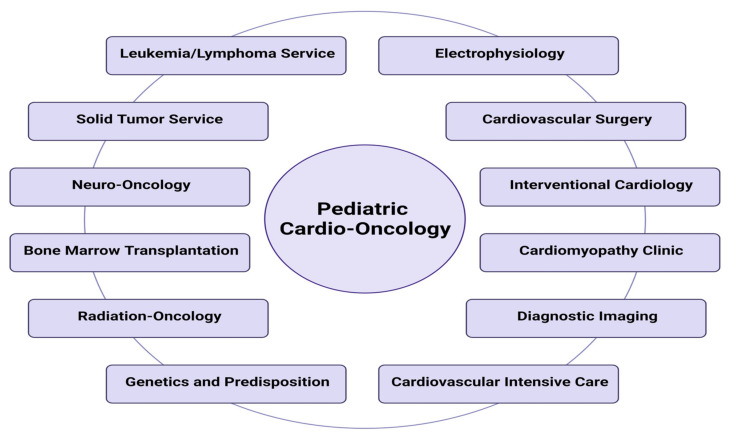
Stakeholders caring for pediatric cardio-oncology patients. Created with BioRender.com.

**Figure 2 children-08-01200-f002:**
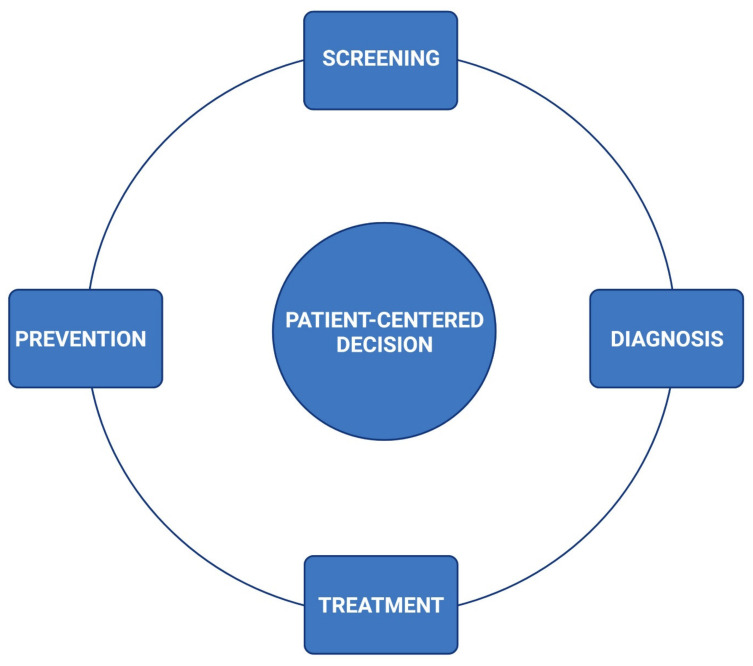
Conceptual model of a continuum of cardiovascular care for children undergoing oncology therapies. Created with BioRender.com.

**Figure 3 children-08-01200-f003:**
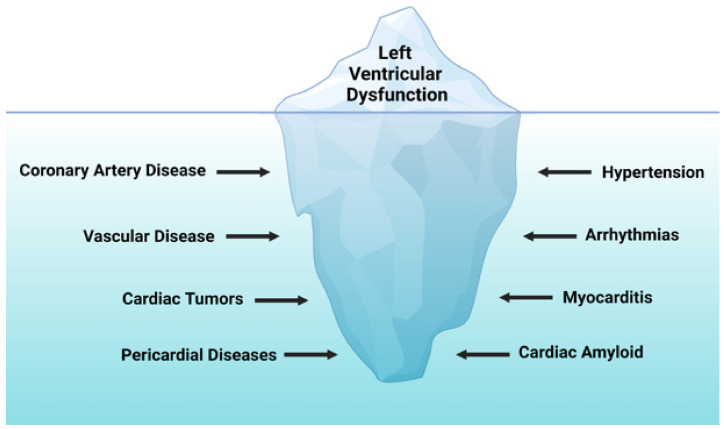
Underlying causes of heart failure in PedCO. Created with BioRender.com.

**Figure 4 children-08-01200-f004:**
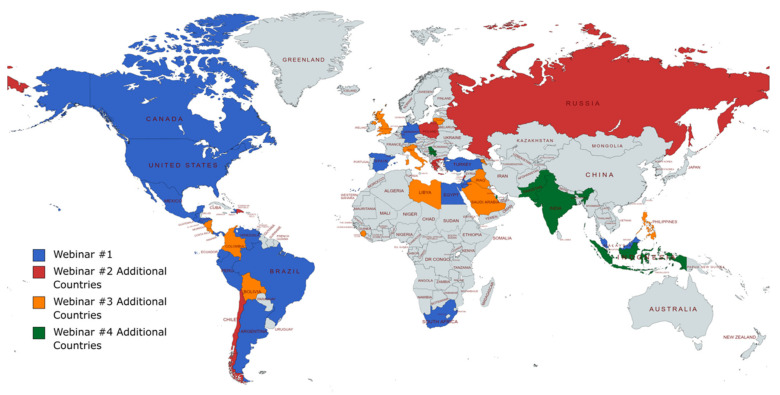
Geographical representation of the LBCH/SJCRH Pediatric Cardio-Oncology webinar series.

**Table 1 children-08-01200-t001:** Common anticancer therapies associated with cardiovascular toxicity [9,10,11,12,13,14].

Types of Anticancer Therapy	Examples Used for Pediatric Cancers	Cardiovascular Toxicities
Alkylating agents	Cyclophosphamide	Arrhythmias Endothelial dysfunction Pericardial effusions Thrombosis
Anthracyclines	Doxorubicin Daunorubicin	Dysrhythmias Endothelial dysfunction Cardiomyopathy (acute, usually reversible) Cardiomyopathy (chronic, usually non-reversible) Oxidative stress
Antimetabolites	Cisplatin 5-Fluorouracil	Dysrhythmias Myocardial ischemia
Immune-based therapies	Immune checkpoint inhibitors Chimeric antigen receptor T-cell therapy	Arterial hypertension Cardiomyopathy (acute, usually reversible) Cardiomyopathy (chronic, usually non-reversible) Cytokine release syndrome Dysrhythmias Endothelial dysfunction Pericardial effusions QTc prolongation Thrombosis
Radiation therapies	Proton radiation Photon radiation	Arterial hypertension Increase pulmonary vasoreactivity Cardiomyopathy (acute, usually reversible) Cardiomyopathy (chronic, usually non-reversible) Pericarditis
Tyrosine kinase inhibitors	Pazopanib Trametinib Sorafenib	Arterial hypertension Cardiomyopathy (acute, usually reversible) Dysrhythmias Endothelial dysfunction Pericardial effusions QTc prolongation Thrombosis
Vinca alkaloids	Vincristine Vinblastine	Myocardial ischemia

**Table 2 children-08-01200-t002:** Most frequent reasons to consult the cardiology team at St. Jude Children’s Research Hospital between 2017–2018.

Primary Reason for Consultation	Percentage Representation
Diastolic dysfunction	31.8%
Dysrhythmias	15.6%
Systolic dysfunction	12.4%
Systemic arterial hypertension	10.1%
Pericardial disease	9.6%
Thromboembolic phenomena	8.9%
Pulmonary arterial hypertension	5.6%

**Table 3 children-08-01200-t003:** Functional classification to diagnose and manage pediatric cardiology-oncology patients.

PedCO Stage	Description	PedCO Characteristics	Therapeutic Options
A	Patients at high risk to develop cardiovascular (CV) toxicity	-Anticancer therapy exposure without signs of pathologic cardiac remodeling or vascular toxicity -Patients scheduled to receive anticancer modalities associated with CV injury -Personal history of CV disease (e.g., diabetes, dyslipidemia, carriers of pathogenic gene variants associated with CV disease)	Primary prevention includes: -Encouraging regular exercise -Management of dyslipidemia, diabetes, and physical deconditioning -Avoidance of alcohol, illicit drugs, and smoking
B	Patients manifesting CV toxicity with no symptoms of heart failure	-Patients with subclinical systolic dysfunction (by ejection fraction or strain analysis), diastolic dysfunction, systemic or pulmonary hypertension, or abnormally elevated cardiac biomarkers	-Include primary prevention recommendations under stage A -Secondary prevention, includes the institution of medical therapy for CTRCD
C	Patients manifesting symptoms of CV toxicity	-Patients with symptoms associated with cancer therapy-related cardiovascular dysfunction	-Include recommendations under stage A and B in addition to managing symptoms of CTRCD (inpatient or outpatient)
D	Advanced CV disease requiring hospital-based support	-Patients with cancer therapy-related CV dysfunction requiring hospital-based support	-Include recommendations under stages A, B, and C -Escalation of care to hospital/intensive care

## Data Availability

No new data were created or analyzed in this study. Data sharing does not apply to this article.

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
