# Peer review of "Pediatric Cardio-Oncology Medicine: A New Approach in Cardiovascular Care"

_children, 2021, doi:10.3390/children8121200_

Round 1
Reviewer 1 Report
this review by Martinez H.R. and collaborators on pediatric cardio oncology tackle a very current complex topic with clarity and with a good updated bibliographic support. Reading is pleasant and the information you receive is useful. I have no objections to raise
Author Response
Thank you for giving me the opportunity to submit a revised draft of my manuscript titled “Pediatric Cardio-Oncology Medicine: A New Approach in Cardiovascular Care” to Children, section: Oncology and Hematology. We appreciate the time and effort that you and the reviewers have dedicated to providing your valuable feedback on my manuscript. We are grateful to the reviewers for their insightful comments on my paper. We have been able to incorporate changes to reflect most of the suggestions provided by the reviewers. We have highlighted the changes within the manuscript. Here is a point-by-point response to the reviewers’ comments and concerns.
Comments from Reviewer 1
Comment 1: This review by Martinez H.R. and collaborators on pediatric cardio oncology tackle a very current complex topic with clarity and with a good updated bibliographic support. Reading is pleasant and the information you receive is useful. I have no objections to raise.
Response 1: Thank you for your comment, we completely agree this is an important topic. Regarding your suggestion about a spelling check, we acquired editing services for professional language polishing and proofreading.
Reviewer 2 Report
This review article stems for the limited availability of specific guidelines in pediatric cardio-oncology (PeCO) and describes the experience of the Heart Institute at Le Bonheur Children’s Hospital to develop multidisciplinary independent clinical pathways to improve the outcome in pediatric oncological subjects, through screening, preventive, monitoring and treatment strategies.
The topic is of current interest, although some major issues should be carefully considered by the authors:
- The manuscript is extensive but lascks some practical scenarios. I invite the authors to consider a clinical vignette/case study with some challenges due to complexity/multiple cardiovascular issues (how a real case was effectively prevented/managed) to describe how PedCo effectively works, thus making easier for a reader to appreciate its clinical utility.
- I think that the manuscript could benefit from providing a table on some specific anticancer drugs used in pediatrics that have been associated with cardio-toxicity. The spectrum of cardiovascular toxicities should be also specified (e.g., arrhythmia, myocarditis, reversible/irreversible heart failure, venous thromboembolism, hypertension). This would offer an additional perspective to the reader.
- The acronym CTRCD is unclear (line 41). Please clarify.
- Table 1 could benefit from adding a column with key notes/comments for the reader, for instance on whether or not the relevant guideline was specifically developed for pediatric subjects, some key recommendations on diagnosis, prevention, management…
- In Table 2, I would consider also providing absolute number (in addition to percentages), and examples of relevant anticancer drugs.
- Line 261 appear to be duplicate.
- With regard to drugs used to treat LVEF (ACEI, ARB, ARNI), some details should be provided in terms of safety issues (with monitoring strategies) and potential dosage adjustment in pediatrics.
- With regard to QT prolongation and Torsade de Pointes, the authors should discuss the role of crediblemeds.org, where a specific section called OncoSupport was recently implemented (https://www.crediblemeds.org/oncosupport)
Author Response
Thank you for giving me the opportunity to submit a revised draft of my manuscript titled “Pediatric Cardio-Oncology Medicine: A New Approach in Cardiovascular Care” to Children, section: Oncology and Hematology. We appreciate the time and effort that you and the reviewers have dedicated to providing your valuable feedback on my manuscript. We are grateful to the reviewers for their insightful comments on my paper. We have been able to incorporate changes to reflect most of the suggestions provided by the reviewers. We have highlighted the changes within the manuscript. Here is a point-by-point response to the reviewers’ comments and concerns.
Comments from Reviewer 2:
Comment 1: The manuscript is extensive but lascks some practical scenarios. I invite the authors to consider a clinical vignette/case study with some challenges due to complexity/multiple cardiovascular issues (how a real case was effectively prevented/managed) to describe how PedCo effectively works, thus making easier for a reader to appreciate its clinical utility.
Response 1: Thank you for pointing this out. The reviewer is correct, and we have adjusted in the text a broad but common clinical scenario of patients with systolic dysfunction who require a unique approach to achieve completion of oncologic therapies. Please find this edit in page 10, between lines 304-313.
Comment 2: I think that the manuscript could benefit from providing a table on some specific anticancer drugs used in pediatrics that have been associated with cardio-toxicity. The spectrum of cardiovascular toxicities should be also specified (e.g., arrhythmia, myocarditis, reversible/irreversible heart failure, venous thromboembolism, hypertension). This would offer an additional perspective to the reader.
Response 2: Your feedback is appreciated. We have included a new table in page 5 which includes some of the most common pediatric anticancer therapies known to cause cardiovascular toxicity.
Comment 3: The acronym CTRCD is unclear (line 41). Please clarify.
Response 3: Thank you. The acronym has been clarified.
Comment 4: Table 1 could benefit from adding a column with key notes/comments for the reader, for instance on whether or not the relevant guideline was specifically developed for pediatric subjects, some key recommendations on diagnosis, prevention, management…
Response 4: Thank you for this suggestion. However, we thought that adding a third column with this information may be redundant as the titles of these guidelines seem to be self-explanatory.
Comment 5: In Table 2, I would consider also providing absolute number (in addition to percentages), and examples of relevant anticancer drugs.
Response 5: We think this is an excellent suggestion. However, a complete assessment to incorporate the number of patients and the associated oncology therapies would not be feasible to perform within the time provided to complete this manuscript.
Comment 6: Line 261 appear to be duplicate.
Response 6: Thank you. The duplicated text has been addressed.
Comment 7: With regard to drugs used to treat LVEF (ACEI, ARB, ARNI), some details should be provided in terms of safety issues (with monitoring strategies) and potential dosage adjustment in pediatrics.
Response 7: We have added the suggested content to the manuscript in lines 286-290 regarding the use of beta blocker therapy, and in lines 300-304 for ACEi and ABRs.
Comment 8: With regard to QT prolongation and Torsade de Pointes, the authors should discuss the role of crediblemeds.org, where a specific section called OncoSupport was recently implemented (https://www.crediblemeds.org/oncosupport).
Response 8: Your suggestion has been incorporated in page 12, line 417-420.
Reviewer 3 Report
This review article, addressing key issues of Pediatric Cardiology-Oncology is interesting and should be useful to the journal readers.
A brief comment RE: a personalized approach to the patients/caregivers, who should be “practically educated” about their own medical problems and treated as “partners” in healthcare by their medical providers (e.g., cardio-oncology teams and primary care staff) via active participation in shared-decision making, could be mentioned in conclusion.
A list of abbreviations [at the end] would be helpful.
Also, providing some additional clear & concise educational resources (e.g., brochures, websites, etc.) for primary care providers and patients/caregivers should be useful.
Author Response
Thank you for giving me the opportunity to submit a revised draft of my manuscript titled “Pediatric Cardio-Oncology Medicine: A New Approach in Cardiovascular Care” to Children, section: Oncology and Hematology. We appreciate the time and effort that you and the reviewers have dedicated to providing your valuable feedback on my manuscript. We are grateful to the reviewers for their insightful comments on my paper. We have been able to incorporate changes to reflect most of the suggestions provided by the reviewers. We have highlighted the changes within the manuscript. Here is a point-by-point response to the reviewers’ comments and concerns.
Comments from Reviewer 3:
Comment 1: A brief comment RE: a personalized approach to the patients/caregivers, who should be “practically educated” about their own medical problems and treated as “partners” in healthcare by their medical providers (e.g., cardio-oncology teams and primary care staff) via active participation in shared-decision making, could be mentioned in conclusion.
Response 1: Thank you for pointing this out. We have addressed your valuable suggestion in page 16, lines 623-632 and 636-637.
Comment 2: A list of abbreviations [at the end] would be helpful.
Response 2: The suggestion has been integrated.
Comment 3: Also, providing some additional clear & concise educational resources (e.g., brochures, websites, etc.) for primary care providers and patients/caregivers should be useful.
Response 3: We have added the suggested content to the manuscript. COG (line 532), Crediblemeds (line 417), ICOS (line 586).
Round 2
Reviewer 2 Report
The authors have addressed the majority of issues.
- Please make sure that the acronym CTRCD stands for "cancer therapy-related cardiovascular toxicity" (maybe DYSFUNCTION?)
- Because previous table 2 was not implemented according to the comments, please consider to implement the new table 2 by providing the prevalence/incidence of cardiovascular dysfunction (for instance from prodct label) in order to support the reader with the epidemiological impact
Author Response
COVER LETTER FOR RE-SUBMISSION OF MANUSCRIPT
Date: November 30, 2021
Thank you for allowing me to submit a revised draft of my manuscript titled “Pediatric Cardio-Oncology Medicine: A New Approach in Cardiovascular Care” to Children, section: Oncology and Hematology. We appreciate the time and effort that the second reviewer has dedicated to providing your valuable feedback. We have highlighted the changes within the manuscript.
Here is a point-by-point response to the reviewers’ comments and concerns.
Comments from Reviewer 2:
Comment 1: Please make sure that the acronym CTRCD stands for "cancer therapy-related cardiovascular toxicity" (maybe DYSFUNCTION?)
Response 1: Thank you for pointing this out again. The reviewer is correct, and we have adjusted the acronym CTRCD (cancer therapy-related cardiovascular dysfunction). Please find this change on page 1, line 40.
Comment 2 (from the initial submission): I think that the manuscript could benefit from providing a table on some specific anticancer drugs used in pediatrics that have been associated with cardio-toxicity. The spectrum of cardiovascular toxicities should be also specified (e.g., arrhythmia, myocarditis, reversible/irreversible heart failure, venous thromboembolism, hypertension). This would offer an additional perspective to the reader.
Comment 2 (from the second revision): Because previous table 2 was not implemented according to the comments, please consider to implement the new table 2 by providing the prevalence/incidence of cardiovascular dysfunction (for instance from prodct label) to support the reader with the epidemiological impact
Response: Thank you for your comment, we completely agree that the table was not reflecting your initial suggestion. During the previous submission, there was a technical error in the way the table was incorporated and the visualization of it was off-site. Please find the edited table on page 5, line 145.
Table 2 is entitled “Common anticancer therapies associated with cardiovascular toxicity)”. The first column includes a list of types of therapies used in children with cancer, the second column includes some examples of these therapies, and the last column contains the spectrum of cardiovascular toxicities associated with these therapies. References have been added to it.
Implementing a table with information suggesting the prevalence/incidence of CTRCD from product labels may not necessarily provide the readers with accurate epidemiological information from the entire spectrum of the cardiovascular adversities. For instance, the FDA provides information about the use of anthracyclines and myocardial insufficiency, but it does not provide compelling information regarding the prevalence/incidence of CTRCD in many of the FDA-approved anticancer drugs (>300), the combination of them in cancer protocols, and the prevalence/incidence beyond the scope of myocardial dysfunction.
Sincerely,
Hugo R. Martinez, MD, FAAP, AFACC
Assistant Professor of Pediatric Cardiology
University of Tennessee Health Science Center
The Heart Institute at Le Bonheur Children’s Hospital
49 N. Dunlap Street, 3rd Floor FOB - 358
Memphis, TN 38105
| 901-287-6819 | Fax 901-287-5970 |
hmartinez@uthsc.edu
